

# Is It Feasible to Use a Single Remote Sensing Optical Water Index for Rapid Mapping of Water Resources?

Yuqing Wang[1,2], Yijie Ma[1,2], Tingsong Gong[1,2], Xueyue Liang[1,2], Yaochen Qin[1,2], Haifeng Tian[1,2,3],
Jie Pei[4], Li Wang[5]

[1]Henan International Joint Laboratory of Geospatial Technology, College of Geography and Environmental Science, Henan University, Kaifeng 475004, China
[2]Key Laboratory of Geospatial Technology for the Middle and Lower Yellow River Regions (Henan University), Ministry of Education, Kaifeng 475004, China
[3]Henan Dabieshan National Field Observation and Research Station of Forest Ecosystem, Xinyang 464000, China
[4]School of Geospatial Engineering and Science, Sun Yat-Sen University, Zhuhai 519082, China
[5]State Key Laboratory of Remote Sensing Science, Aerospace Information Research Institute, Chinese Academy of Sciences, Beijing 100101, China

*Correspondence to*: Haifeng Tian (tianhaifeng@henu.edu.cn)

**Abstract.** Water resources are an important component of the earth's system, and the frequent occurrence of floods and droughts in the context of current climate change makes rapid and accurate monitoring of water resources particularly important. The optical water index (OWI) is a commonly used method for extracting water areas on the basis of remote sensing images, often with a high level of automation. However, selecting the right OWI is challenging due to the variety of water types. To quantitatively evaluate the differences in the mapping potential of different OWIs for surface water, we selected 12 commonly used OWIs to conduct comparative experiments among five types of surface water based on Landsat-8 and Sentinel-2 images. The results revealed that the Normalized Difference Water Index (NDWI) was better for turbid water, the Multi-Band Water Index (MBWI) was better for shaded water, the Modified Normalized Difference Water Index (MNDWI) was better for green water, and the Automated Water Extraction Index (AWEI$_{sh}$)was better for swamp water and saltwater. Sentinel-2 has a higher ability to classify water than Landsat-8. Our work provides prior experience for fast and accurate water resources mapping in case of floods or droughts.

## 1 Introduction

Surface water is highly dynamic(Pekel et al., 2016) and at the same time it is an important part of the global ecosystem and climate system (Zhang et al., 2022). Monitoring these changes in surface water marks the initial stage of advancing research on ecosystem and flood management (Dronova et al., 2011). Remote sensing technology is widely regarded as a cost-effective and efficient method for monitoring large-scale water resources(Huang et al., 2018). Compared with machine learning methods, the optical water index (OWI) method presents notable advantages in simplicity, efficiency, and repeatability, rendering it





highly suitable for long-term and large-scale analyses across diverse environmental applications (Du et al., 2014), and it is also widely used in hazard mapping in conjunction with SAR because the most serious problem with optical imaging is the influence of clouds (Psomiadis et al., 2020). Wang et al. (2019) used a method of extracting water by combining OWIs and vegetation

indexes to monitor the changes in inundated hydrological conditions in Poyang Lake before and after the completion of the Three Gorges Dam from 1988 to 2016. Zhang et al. (2022) used a method of combining water index to monitor the changes in open water in the Yellow River Basin from 1986 to 2020. Luying et al. (2023) used the MNDWI to study the relationship between surface water change and climate change in Qinghai Province from 1986 to 2018.

Classification methods that use OWIs as inputs to classify water and other classes from OWIs can be roughly divided into

three categories: supervised classification, unsupervised classification, and threshold methods (Li et al., 2016). Some scholars use the threshold method (Adrian et al., 2016); however, choosing the optimal threshold is very complex and time-consuming (Liu et al., 2023). The Otsu method (1979) is a widely used automatic thresholding method aimed at maximizing interclass variance and minimizing intraclass variance (Du et al., 2016). However, the algorithm does not yield good results for images without bimodal features (Zhou et al., 2015). Supervised classification and its accuracy is depend on the quality of the training

samples (Shin et al., 2016). An unsupervised classification algorithm does not rely on training samples and has less subjective interference (Tian et al., 2024). Therefore, we ultimately chose an unsupervised classification method to classify OWI images to pursue objective evaluation results.

In fact, as sensors progress, the OWI develops (Huang et al., 2018). Most OWI methods are constructed on the basis of Landsat series images. Although MODIS data are frequently used for surface monitoring, their coarse spatial resolution often

leads to suboptimal classification accuracy for mixed water and non-water pixels(Tian et al., 2017). The emergence of Sentinel-2 has significantly enhanced the monitoring of water areas via remote sensing owing to its higher spatial resolution and increased spectral bands(Zeng et al., 2022).

The concept of OWIs can be traced back to 1985, when tasseled cap transformations were primarily used. Specifically, Tasseled Cap Wetness (TCW) offers features closely related to the wetness features derived from Thematic Mapper (TM)

Tasseled Cap transformations (Crist, 1985). Eleven years later, the Normalized Difference Water Index (NDWI) was introduced, inspired by the Normalized Difference Vegetation Index (NDVI). The NDWI capitalizes on the fact that water exhibits higher reflectance in the green band than in the near-infrared band, whereas soil and vegetation typically show the opposite pattern. Consequently, water features yield positive values and are accentuated, whereas vegetation and soil tend to yield zero or negative values and are subdued (Mcfeeters, 1996). The Normalized Difference Water Index 3($NDWI_3$) method

was subsequently introduced to address the challenge of accurately delineating the transitional zone between water and non-water regions in TCW images. Ouma and Tateishi (2006) aimed to increase the precision of water boundary extraction, and they used the bands in the TCW images and the normalized structure of the NDWI, with the $NDWI_3$ being the most accurate.

In the same year, Xu (2006) introduced refinements to minimize the effects of intervening in the building background, substituting the mid-infrared band for the near-infrared band in the NDWI to construct the Modified Normalized Difference

Water Index (MNDWI). One year later, Yan et al. (2007) considered Henan Province, China, an arid and semiarid region with



a complex background environment that included vegetation, cultivated land, buildings, and dry river channels and that a single band difference would not be sufficient to eliminate all sources of interference; thus, the Enhanced Water Index (EWI), which combines the, was introduced. NDWI and MNDWI. This is the first study in which noticed the effects of a dry river on watershed extraction have been reported. Feng (2012) subsequently added a second shortwave infrared band using Landsat
imagery to construct the New Water Index (NWI).

The Automated Water Extraction Index (AWEI) is meticulously crafted to increase robustness and precision while maintaining a stable threshold. Within its design, the iterative process establishes a stability threshold, denoted 0, crucial for discerning water from non-water areas. Comprising two distinct formulas, $AWEI_{nsh}$ and $AWEI_{sh}$, the latter is specifically engineered to increase accuracy by eliminating shadow pixels that $AWEI_{nsh}$ may not effectively address (Feyisa et al., 2014).
To enhance its applicability across diverse environmental backgrounds, Index of Water Surfaces (IWS) employs two division operations to adjust the image to only two gray levels, resulting in a soil brilliance that resembles that of other land uses except for water (Hassani et al., 2015). Notably, in this article on the construction of the IWS, classification results with those of other OWIs are not compared. The proposed Water Index 2015 ($WI_{2015}$) discriminates surface reflectance into distinct land classes linearly and determines the coefficients of minimum intraclass variance and maximum interclass variance (Adrian
et al., 2016). This approach was adopted to increase the level of automation so that the threshold can still have good accuracy when it varies over a small range of values. Later, Wang et al. (2018) proposed the Multi-Band Water Index (MBWI) due to the misclassification of low-reflectivity surfaces because of their similarity in reflectivity to water.

After the launch of Sentinel-2, its unique vegetation-sensitive red-edge band was introduced to construct the Sentinel-2 Water Index (SWI) (Jiang et al., 2021).

**Table 1. Formula for the optical water index**.

| Index | Equation | Reference |
|---|---|---|
| TCW | $TCW = 0.1509 \times \beta_{blue} + 0.1973 \times \beta_{green} + 0.3279 \times \beta_{red}$ $+ 0.3406 \times \beta_{nir} - 0.7112 \times \beta_{swir1} - 0.4572 \times \beta_{swir2}$   (1) | (Crist, 1985) |
| NDWI | $NDWI = \dfrac{\beta_{green} - \beta_{nir}}{\beta_{green} + \beta_{nir}}$   (2) | (Mcfeeters, 1996) |
| NDWI$_3$ | $NDWI_3 = \dfrac{\beta_{nir} - \beta_{swir1}}{\beta_{nir} + \beta_{swir1}}$   (3) | (Ouma and Tateishi, 2006) |
| MNDWI | $MNDWI = \dfrac{\beta_{green} - \beta_{swir1}}{\beta_{green} + \beta_{swir1}}$   (4) | (Xu, 2006) |





| EWI | $$EWI = \frac{\beta_{green} - \beta_{nir} - \beta_{swir1}}{\beta_{green} + \beta_{nir} + \beta_{swir1}} \quad (5)$$ | (Yan et al., 2007) |
|---|---|---|
| NWI | $$NWI = \frac{\beta_{blue} - \beta_{nir} - \beta_{swir1} - \beta_{swir2}}{\beta_{blue} + \beta_{nir} + \beta_{swir1} + \beta_{swir2}} \quad (6)$$ | (Feng, 2012) |
| AWEI | $$AWEI_{nsh} = 4 \times (\beta_{green} - \beta_{swir1}) - 0.25 \times \beta_{nir} + 2.75 \times \beta_{swir2} \quad (7)$$ $$AWEI_{sh} = \beta_{blue} + 2.5 \times \beta_{green} - 1.5 \times (\beta_{nir} + \beta_{swir1}) - 0.25 \times \beta_{swir2} \quad (8)$$ | (Feyisa et al., 2014) |
| IWS | $$IWS = \frac{2 \times (4 \times \beta_{swir1} - \beta_{blue})}{\beta_{swir1}} - 2 \times \frac{\beta_{swir1}}{\beta_{blue}} \quad (9)$$ | (Hassani et al., 2015) |
| WI$_{2015}$ | $$WI_{2015} = 1.7204 + 171 \times \beta_{blue} + 3 \times \beta_{green} - 70 \times \beta_{red} - 45 \times \beta_{nir} - 71_{swir2} \quad (10)$$ | (Adrian et al., 2016) |
| MBWI | $$MBWI = 2 \times \beta_{green} - \beta_{red} - \beta_{nir} - \beta_{swir1} - \beta_{swir2} \quad (11)$$ | (Wang et al., 2018) |
| SWI | $$SWI = \frac{\beta_{vre1} - \beta_{swir2}}{\beta_{vre1} + \beta_{swir2}} \quad (12)$$ | (Jiang et al., 2021) |

Table **1** shows the formula for each OWI, where $\beta_{blue}$ represents the reflectance of the blue band (B2 of Landsat-8 and Sentinel-2), $\beta_{green}$ represents the reflectance of the green band (B3 of Landsat-8 and Sentinel-2), $\beta_{red}$ represents the reflectance of the red band (B4 of Landsat-8 and Sentinel-2), $\beta_{nir}$ represents the reflectance of the near-infrared band (B5 of Landsat-8, B8 of Sentinel-2), $\beta_{swir1}$ represents the reflectance of the first shortwave infrared band (B6 of Landsat-8, B11 of Sentinel-2) and $\beta_{swir2}$ represents the reflectance of the second shortwave infrared band (B7 of Landsat-8, B12 of Sentinel-2), and $\beta_{vre1}$ represents the reflectance of the vegetation red edge 1 (B5 of Sentinel-2).

Although many OWIs are available, they can be deceptive. Because of the variety of types of water, including spectral variations and their presence in various background environments, it is not feasible to simply apply one formula to all conditions (Zeng et al., 2022). However, currently, comprehensive comparative analyses of OWIs that demonstrate the strengths and weaknesses of different indices across various types of water and background environments are lacking. Research on the overall performance of these indicators in different types of water and complex and variable environments in different regions of the world is lacking. No single OWI stands out as the best, as each index has different applicability. In this work, we aim to quantify the advantages and disadvantages of twelve OWIs. In this work, we aim to quantify the strengths and weaknesses of the twelve OWIs, which can provide technical and theoretical support for better water resources monitoring and rapid mapping in case of related disasters.



## 2 Materials and methods

### 2.1 Study area

As shown in Figure 1., in this study, we selected a total of 10 study areas through a literature review as well as field surveys,

namely, the Yellow River, the Nile River, Lake Taihu, the Danube River, Namtso Lake, Lake Eyre, Lake Geneva, the Charles River, Poyang Lake, and Lake Okeechobee. The 10 study areas comprehensively represent five types of water in the world, namely, turbid water, green water, saltwater, water in shadow, and swamp water, which are suitable for studying the effects of different OWIs.

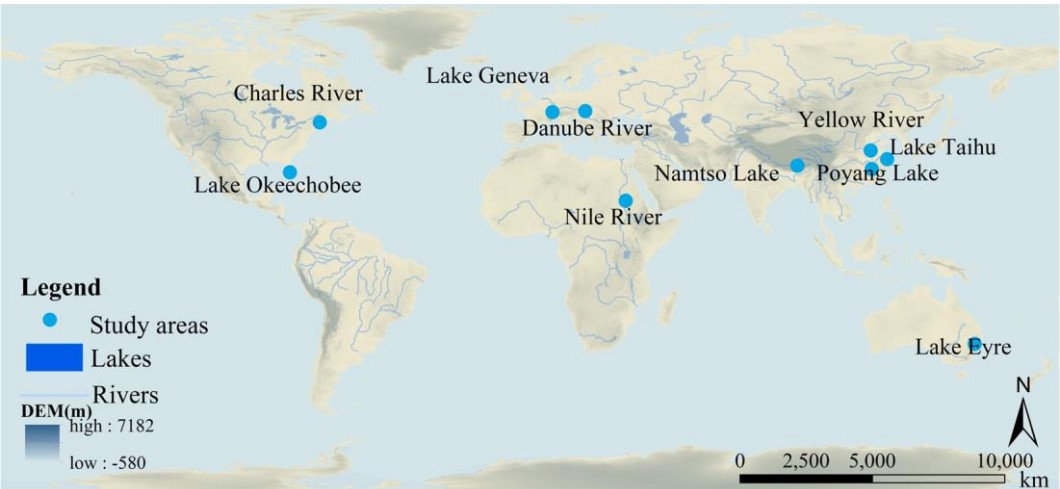

**Figure 1. Distribution of the study area locations. DEM comes from ENVI 5.3 software publicly available data.**

The Yellow River is the second-largest river in China and is known for its high sediment content (Shen et al., 2010). The Blue Nile, a tributary of the Nile River, carries two-thirds of the sediment of the world's longest river(Janse Van Vuuren et al., 2018).

The Hungarian segment of the Danube River and Taihu Lake exemplify eutrophic waters, and we chose these two research

areas to study green water. Lake Taihu is located in one of the world's most heavily populated regions, and a large amount of polluted water is discharged into Lake Taihu, resulting in a phytoplankton bloom (Le et al., 2009). At present, the water bloom in Taihu Lake is still serious(Wang et al., 2020). The Danube River is the most international river in the world (Sommerwerk et al., 2010) and Europe's second-largest river. Blooms have also been observed in the Danube's waters, and water quality is a major concern (Horvat et al., 2021).

Namtso Lake and Lake Eyre are both saltwater lakes, so we selected these two study areas to assess the effectiveness of OWIs for saltwater. Namtso Lake, which is located on the Qinghai-Tibet Platea, is the highest lake in the world and the second-largest inland saltwater lake in China (Xu and Kang, 2010). Lake Eyre is the largest salt lake and lowest point in Australia (Nanson and Price, 1998).





Lake Geneva and the Charles River, one surrounded by mountains and the other by buildings, were used to investigate
the impact of OWI extraction in shaded regions. Lake Geneva, situated in Switzerland, is encircled by the Alps, the largest
mountain range in Western Europe, and it is a mountain-surrounded lake (Lemmin et al., 2005). The Charles River originates
in Newton and flows through a linear formation of university campuses and affiliated R&D complexes, ultimately terminating
in Boston(Gahagan and Bailey, 2020).

We selected the wetland ecosystems of Poyang Lake and Lake Okeechobee as our study sites to investigate the efficacy
of OWIs in differentiating swamps from surface waters. Poyang Lake, China's largest freshwater lake, experiences significant
fluctuations in water area (Guiping et al., 2014; Tian et al., 2017). The Poyang Lake wetland, which is situated in the littoral
zone of the lake, is among the earliest sites designated Wetlands of International Importance under the Ramsar Convention
(Secretariat, 2010). Lake Okeechobee, situated at the heart of the broader Kissimmee River Lake Okeechobee Everglades
ecosystem in South Florida, plays a pivotal role in a range of ecosystem and water management functions(Gahagan and Bailey,
135  2020).

## 2.2 Data

Google Earth Engine (GEE), which stores a large number of pre-processed remotely sensed images(Gorelick et al., 2017), we
used for surface reflectance (SR) data. The spatial resolution of OWIs computed by Landsat-8 is 30m and that of OWIs
computed by Sentinel-2 is 10m. The data used in this study are shown in Table 2. In addition to remotely sensed imagery data,
we also combined ground-truth observation data.

**Table 2: Type of satellite, type of water, time of image acquisition, and location.**

| Satellite Type | Water Type | Date Acquisition | Study Area |
|---|---|---|---|
| Landsat-8 | Turbid water | February 9, 2020 | Yellow River |
| | | September 21, 2021 | Nile River |
| | Green water | July 28, 2021 | Danube River |
| | | June 23, 2021 | Taihu Lake |
| | Salt water | November 3, 2021 | Namtso Lake |
| | | May 19, 2021 | Lake Eyre |
| | Shaded water | July 29, 2021 | Lake Geneva |
| | | July 27, 2021 | Charles River |
| | Swamp water | January 12, 2021 | Poyang Lake |
| | | June 24, 2022 | Lake Okeechobee |
| Sentinel-2A | Turbid water | March 3, 2021 | Yellow River |
| | | February 1, 2022 | Nile River |
| | Green water | June 21, 2021 | Danube River |





| | June 23, 2021 | Taihu Lake |
|---|---|---|
| Salt water | October 05, 2021 | Namtso Lake |
| | April 30, 2021 | Lake Eyre |
| Shaded water | May 31, 2021 | Lake Geneva |
| | May 27, 2021 | Charles River |
| Swamp water | January 1, 2021 | Poyang Lake |
| | September 30, 2022 | Lake Okeechobee |

## 2.3 Method

The K-means classification method is widely employed for unsupervised classification. In the K-means algorithm, cluster analysis is utilized to randomly determine the central locations of clusters and subsequently group the objects that are closest

to these centers (Piloyan and Konečný, 2017). Through iterative calculations, the values of each clustering center are updated individually until the optimal clustering outcome is achieved. In this study, the calculation results of twelve OWIs were classified via the K-means method. Specifically, the K-means classification method was employed to partition the images into five categories through ten iterations. Each scene contains approximately five categories: water, vegetation, impervious cover, bare ground, and swamp-and 10 iterations constitute the number of iterations with the highest accuracy that we obtained after

repeated experiments. These categories were then aggregated into two overarching groups, 'water' and 'non-water', and the merging process compared true color images, thereby constructing a binary representation. Here, we classify swamp as non-water. Although it has a high-water content, it does not have fluidity.

The performance of these OWIs in unclear water varies significantly, primarily due to spectral variations. We quantified the spectral variations in different water types by calculating the percentage increase or decrease in various bands across five

distinct categories, namely turbid water, green water, shaded water, swamp water, and saltwater, relative to that for clear reference water. The calculation formula is as follows:

$$\delta = \frac{\beta_{unclear} - \beta_{clear}}{\beta_{clear}} \times 100\%$$

(13)

where $\delta$ represents the percentage increase or decrease in each band for different types of water, $\beta_{unclear}$ denotes the

reflectance of each band of the unclear water, and $\beta_{clear}$ signifies the reflectance of each spectral band of water in the first-class water source protection area. Spectral samples of clear water were obtained from the Danjiangkou Reservoir, which is the largest artificial freshwater lake in Asia and a national first-class water source protection area (Pan et al., 2021). Spectral samples of the Danjiangkou Reservoir were obtained from Landsat-8 imagery from March 2, 2017, and September 21, 2021, and we used these spectral samples as benchmarks to quantify the degree of spectral variability in several other types of water.


The images from these two days are cloudless and the water quality is good. The spectral data of the remaining five water

types and various background features were acquired from 10 study areas via the Landsat-8 dataset (Figure 2.).

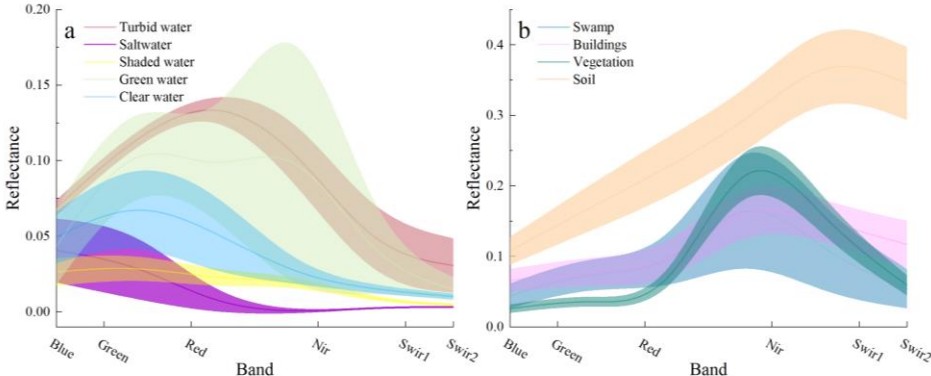

**Figure 2: (a) Spectral curves for 5 types of water. (b) Spectral curves from swamps, buildings, vegetation, and soil.**

We first validated the accuracy of the 12 OWIs, a step based on the average of the accuracies of two types of satellite

imagery, Landsat-8, and Sentinel-2, in 10 study areas for five different types of water, with the exception of the SWI, for which

the accuracy validation was based on only one type of satellite imagery, Sentinel-2, because the $\beta_{vre1}$ used is unique to Sentinel-

2. We used this method to reduce the difference in accuracy among different sensors. In the second step, we explain the reasons

for the correct and incorrect identification of 11 OWIs based on spectra and structures that can be calculated based on Landsat

images. In the third step, we compared the difference of water recognition accuracy between Landsat-8 and Sentinel-2. The

overall process is shown in the following Figure 3.


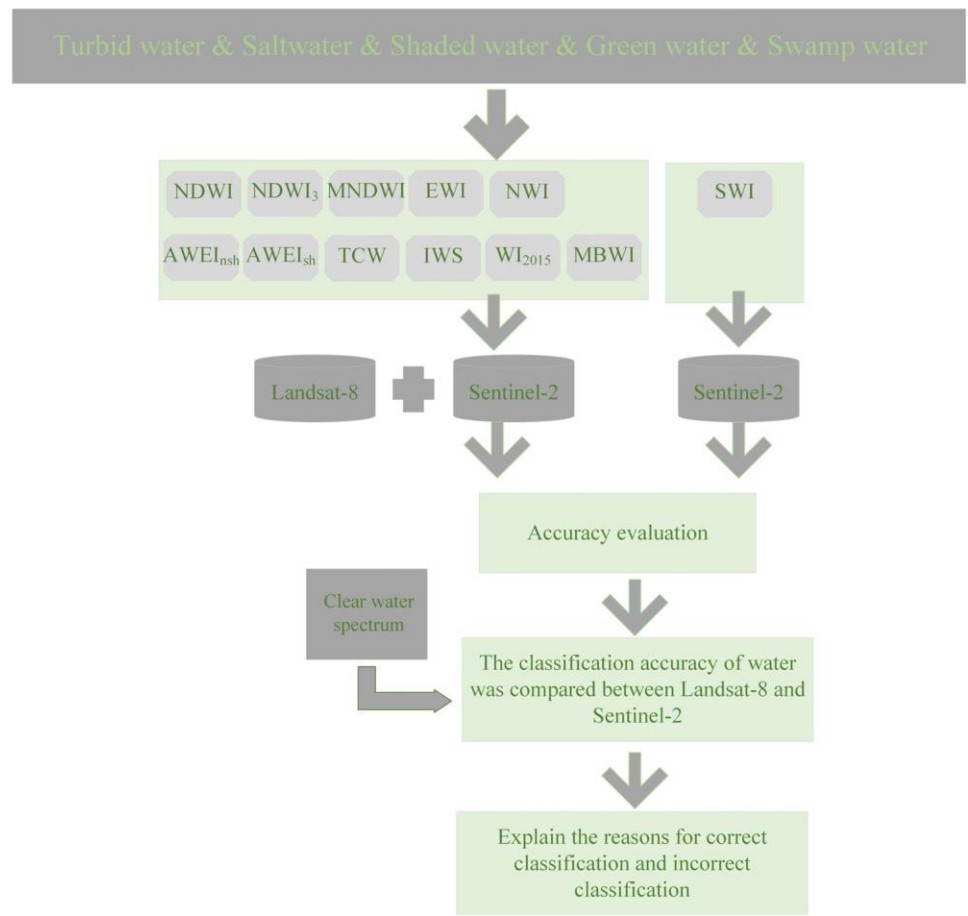

**Figure 3: Study flow chart.**

As accuracy verification results can be significantly influenced by the selection of validation samples (Adrian et al., 2016), we adopted a systematic approach by selecting 100 sample quadrats, each measuring 6 km×6 km. These quadrats covered the

interface between aquatic and terrestrial environments within each study area. Whether it is water or not was determined through manual digitization of true-color images. Additionally, we employed the confusion matrix accuracy validation method to report the classification accuracy, including the overall accuracy (OA), kappa coefficient (Ka), producer accuracy (PA), and user accuracy (UA). We selected these indicators to show the overall classification level of these OWIs, as well as the mismarking and missing mark scenarios. The specific calculations are as follows:

$$OA = \frac{TP + TN}{TA}$$

$\qquad\qquad\qquad\qquad\qquad\qquad\qquad\qquad\qquad\qquad\qquad\qquad$ (14)

$$Ka = \frac{TP + TN - [(TP + FP) \times (TP + FN) + (FN + TN) \times (FP + TN)]}{TA^2 - [(TP + FP) \times (TP + FN) + (FN + TN) \times (FP + TN)]}$$

$\qquad\qquad\qquad\qquad\qquad\qquad\qquad\qquad\qquad\qquad\qquad\qquad$ (15)





$$PA = \frac{TP}{TP + FN} \tag{16}$$

$$UA = \frac{TP}{TP + FP} \tag{17}$$

where TA denotes the overall sample size, TP denotes the total number of image elements correctly identified as water,
TN denotes the total number of non-water image elements correctly classified as such, FP denotes the total number of image
elements incorrectly classified as water, and FN denotes the total number of image elements incorrectly classified as non-
water.

## 3 Results

Establishing a priori knowledge of the recognition accuracy of different water indexes in different situations is essential for
rapid automated mapping in the event of a disaster. Waters of different types exhibit distinct spectral characteristics (Figure
2.), and clear water is markedly different from the four other types of water, resulting in the varying performance of OWIs for
different water types. The image below (Figure 4.) shows 12 OWIs used to extract the average accuracy of five different types
of water.

The extraction results for swamp water demonstrated overall accuracies ranging from 82.56% to 95.15%, with kappa
coefficients ranging from 0.65 to 0.91. Notably, the AWEI$_{sh}$ and NDWI show superior discriminatory capabilities for swamp
water, achieving overall accuracies of 95.15% and 94.81%, respectively, with corresponding kappa coefficients of 0.90 and
0.91. Similarly, was the EWI, had an overall accuracy of 94.61% and a kappa coefficient of 0.91. Conversely, WI$_{2015}$ and the
TCW exhibited lower performances in terms of swamp water extraction, with overall accuracies of 82.56% and 85.59% and
kappa coefficients of 0.65 and 0.74, respectively.

Overall, the extraction accuracy of the twelve OWIs for shaded water was lower than that of the other four types of water,
with an average overall accuracy of 85.08%, ranging from 60.63% to 96.19%, and kappa coefficients ranging from 0.40 to
0.86. The highest accuracy was achieved by the MBWI, with an average overall accuracy of 96.19%, followed closely by the
MNDWI, with 96.13% accuracy. The MBWI also had the highest kappa coefficient of 0.86, followed by the MNDWI and
WI$_{2015}$ with a kappa coefficient of 0.82. Overall, the MBWI demonstrated the most effective separation of shaded water.

For saltwater, the AWEI$_{sh}$ and MBWI demonstrated high overall accuracies of 97.91% and 97.56%, respectively, with
matching kappa coefficients of 0.96. The NDWI followed closely as the third-best performer, achieving an overall accuracy
of 96.61% and a kappa coefficient of 0.95. In contrast, NDWI$_3$ exhibited the least effective extraction for saltwater, with an
overall accuracy of 79.58% and a kappa coefficient of 0.70. Overall, the AWEI$_{sh}$ is generally considered superior in extraction
for saltwater.





The extraction performance of different OWIs varied for green water, with overall accuracies ranging from 74.36% to 97.1% and kappa coefficients ranging from 0.50 to 0.94. The MNDWI and SWI achieved overall accuracies higher than 95.00%, with the MNDWI exhibiting the highest accuracy. Conversely, $NDWI_3$, the IWS, and $WI_{2015}$, with overall accuracies lower than 85%, also had kappa coefficients below 0.70. Among these methods, the IWS had the poorest separation effect for green water, with an overall accuracy of 74.36% and a kappa coefficient of 0.50.

For turbid water, the NDWI demonstrated the most effective extraction, achieving an overall accuracy of 95.20% and a kappa coefficient of 0.89. The EWI and NWI followed closely, with overall accuracies of 93.59% and 91.62%, respectively. In contrast, $WI_{2015}$ had the poorest performance in separating turbid water, with an overall accuracy of 75.37% and a kappa coefficient of 0.59.

     Among the 12 OWIs, the MNDWI and MBWI displayed the highest stability for different types of water, with mean
overall accuracy values of 94.63% and 93.33% and a mean kappa coefficient value of 0.87. Conversely, $NDWI_3$, the IWS, and $WI_{2015}$ exhibited the lowest stability in accurately extracting all types of water, with average overall accuracies of 77.33%, 83.28%, and 85.78%, respectively, and average kappa coefficients of 0.63, 0.70, and 0.72, respectively. Notably, the $AWEI_{sh}$ demonstrated a better separation effect for various types of water than the $AWEI_{nsh}$ did, confirming the findings of previous studies (Zhang et al., 2017).

The TCW, the $AWEI_{nsh}$, the $AWEI_{sh}$, and $WI_{2015}$ were established via iterative or regression methods. The average overall accuracy and kappa coefficient were 87.57% and 0.78, respectively, both of which were lower than the 12 OWIs (89.02% and 0.79, respectively). Among the 12 OWIs, $NDWI_3$ exhibited the most unstable downward trend from the near-infrared band to the shortwave infrared band. Upon excluding this index, the average accuracy of the remaining 11 water indexes increased to 90.08%, with a kappa coefficient of 0.81.

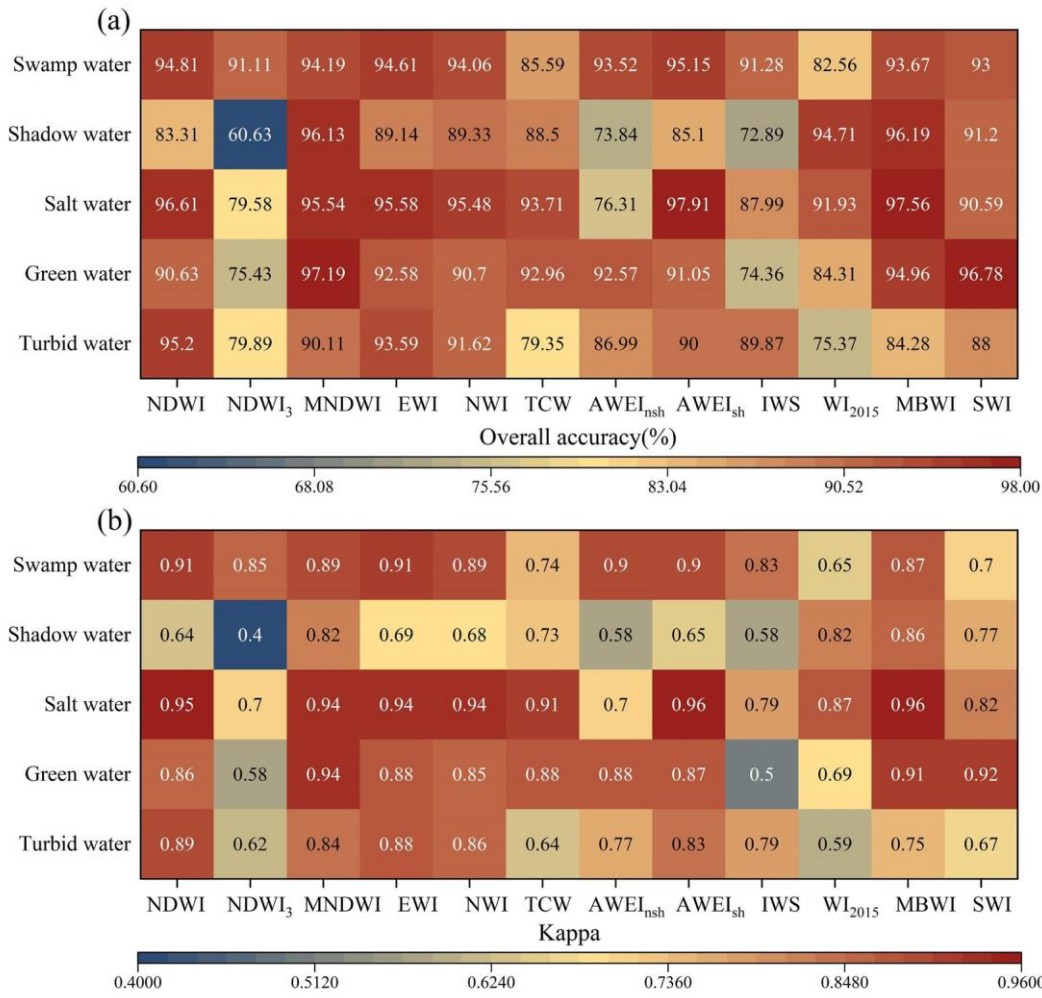

**Figure 4: The mean values of the 12 optical water indexes in the extraction results are indicative of both (a) the overall accuracy and (b) the kappa coefficient for the five types of water.**

The average user accuracy and producer accuracy values for the extraction of various waters were computed for the 12 OWIs (Figure 5.). For turbidity water, all 12 OWIs, with the exception of the IWS, exhibit a producer accuracy greater than the user accuracy. Conversely, for swamp water, except for the SWI and AWEI$_{nsh}$, the user accuracy is greater than the producer accuracy, whereas the remaining 10 OWIs result in the producer accuracy exceeding the user accuracy. Similarly, for shaded water, all 12 OWIs exhibit a producer accuracy that surpasses the user accuracy. This suggests that for the three types of water, misclassification was more prevalent than leakage was, resulting in numerous non-water objects being erroneously classified as water. For green water, excluding the TCW and WI$_{2015}$, and for saltwater, excluding the TCW and AWEI$_{nsh}$, these exceptions are where the producer accuracy exceeds the user accuracy. Conversely, among the remaining 10 OWIs for these two types, the user accuracy exceeds the producer accuracy. This implies a greater number of missing pixels than the number of instances




where water is classified as other features. The reasons for the specific misclassifications and omissions are explained in more detail in the discussion section.

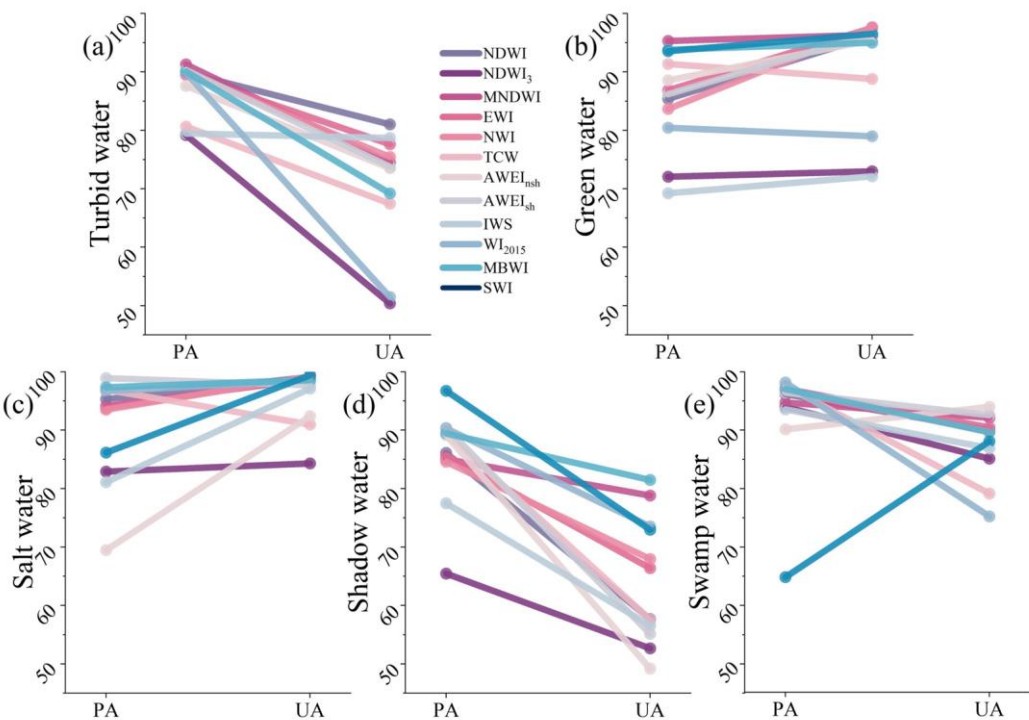

**Figure 5: Comparison of producer accuracy and user accuracy for the 12 OWIs for five different types of water. PA = producer accuracy; UA = user accuracy.**

Choosing the right sensor is also critical for more accurate water monitoring. We compared all Landsat-8 and Sentinel-2 results calculated in this work to evaluate the performance of the two sensors in water recognition (Figure 6.). The overall accuracy of Sentinel-2 and Landsat-8 was 89.16% and 88.60%, respectively. Kappa coefficients were 0.75 and 0.71, respectively. In general, Sentinel-2 sensor is slightly better than Landsat-8 in separating water. This result also confirms previous studies (Zhou et al., 2017).





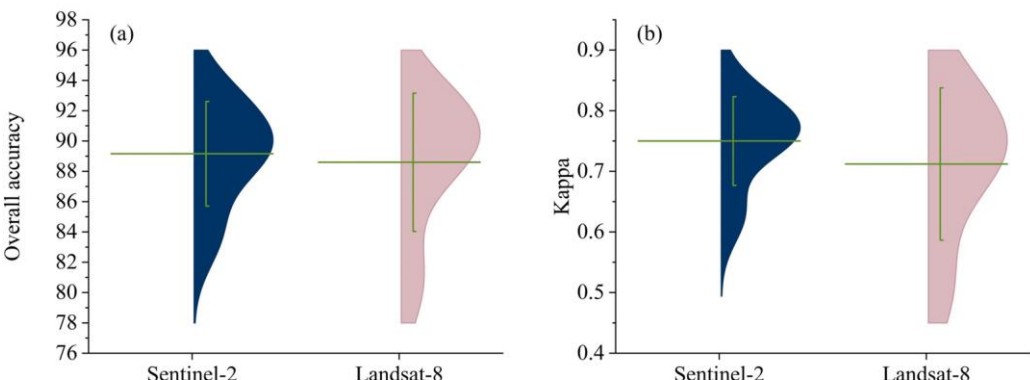

**Figure 6:Comparison of water identification accuracy between Landsat-8 and Sentinel-2. (a) the overall accuracy, (b) the Kappa coefficient.**

## 4 Discussion

### 4.1 Explanation of the phenomenon

Understanding the reasons for the different OWIs identification results is important to improve this highly automated water identification method for better regulation during hydrological hazards. Various OWIs utilize distinct spectral information to enhance water characteristics, resulting in different classification results for the same area. Explaining the reasons behind these divergent classification outcomes of OWIs and summarizing the patterns observed can provide valuable insights for the development of future OWIs.

Specifically, turbid water demonstrates a substantial increase in reflectance across the red, near-infrared, and two shortwave infrared bands. The red band experiences an increase of 169.74%, whereas the near-infrared band exhibits a remarkable increase of 356.00%.

Additionally, the first shortwave infrared band shows a notable increase of 174.41%, and the second shortwave infrared band displays a significant increase of 199.12%. This behavior elucidates the resemblance of turbid water treated by OWIs such as $WI_{2015}$, the MBWI, and the TCW, which may resemble vegetation and swamps (Figure 7.).

For example, when the TCW is employed, both turbid water and swamps are assigned values of 0.02, whereas turbid water treated by the MBWI has a value of -0.28, which closely resembles the swamp's value of -0.27. Similarly, the processing results for turbid water, vegetation, and swamps in the $WI_{2015}$ images are -9.18, -12.56, and -8.80, respectively, while the clear water score is 4.85. Consequently, in images where turbid water, vegetation, and swamps appear to be similar, the confusion experienced by $WI_{2015}$ in extraction becomes more pronounced.



The difference in reflectance between the red wave band and the first and second shortwave infrared bands decreases from 0.03, 0.04, and 0.04 to -0.13, -0.03, and 0.02, respectively. This reduction in differentiation capacity diminished the

index's effectiveness. The normalized values for turbid water in the near-infrared and first shortwave infrared bands, with a value of 0.34, were closely aligned with those for vegetation and swamps, with values of 0.33 and 0.34, respectively. Because $NDWI_3$ relies solely on these two bands, it struggles to discriminate turbid water from the surrounding swamp and vegetation.

The IWS index uses the sum of the ratio of the blue wave band to the first shortwave infrared band and a reciprocal. In the context of turbid water, this ratio is 0.43, with its reciprocal of 2.33. However, this ratio is similar to that observed for soil

(0.27, with a reciprocal of 3.70) and buildings (0.45, with a reciprocal of 2.22).

Compared with that of clear water, the spectral curve of the near-infrared band exhibited a 384.61% increase from the absorption valley to the reflection peak, with a standard deviation of 0.11. The reflectance of the near-infrared band increased from 0.02 to 0.12. Simultaneously, there was a decrease of 12.02% in the blue band and an increase of 58.92% in the green band due to varying algae contents within the waters. In green water, the spectral curve of bloom water can be observed, and

a spectral curve similar to that of clear water can be seen. This is why the NDWI, EWI, $AWEI_{nsh}$, $AWEI_{sh}$, MBWI, and NWI, which are different from the blue or green band and the near-infrared band, all show omissions in the extraction of green water (Figure 7.).

The reflection of the first shortwave infrared band of the green water is still 0.02. The reflectance of this band in clear water is 0.01, which is similar to that of the two types of water. This is why the MNDWI, which does not use the near-infrared

band but uses only the difference between the green band and the shortwave infrared band, can extract water with blooms better. Despite the utilization of six bands, including near-infrared bands, the TCW still accurately identifies green water. Moreover, considering its underlying principles and structural design, the positive coefficient of the TCW for the near-infrared band ensures consistent detection results, even with the increased reflectivity in green waters.

Considering the normalization values of the near-infrared and shortwave infrared bands, which are 0.50 for green water,

0.25 for normal water, 0.33 for vegetation, and 0.34 for swamps, the treatment outcome of green water is more similar to that of vegetation and swamps than to that of normal water. Consequently, $NDWI_3$ classifies green water into one category, with vegetation and swamps while distinguishing it from normal water.

The ratio of the blue band and the first shortwave infrared band of the green water and its reciprocal sum is 2.13, and that of the clear water is 2.70. Because these two ratios are negative, the green water presents a bright color similar to that of other

ground objects in the IWS image and is difficult to distinguish.



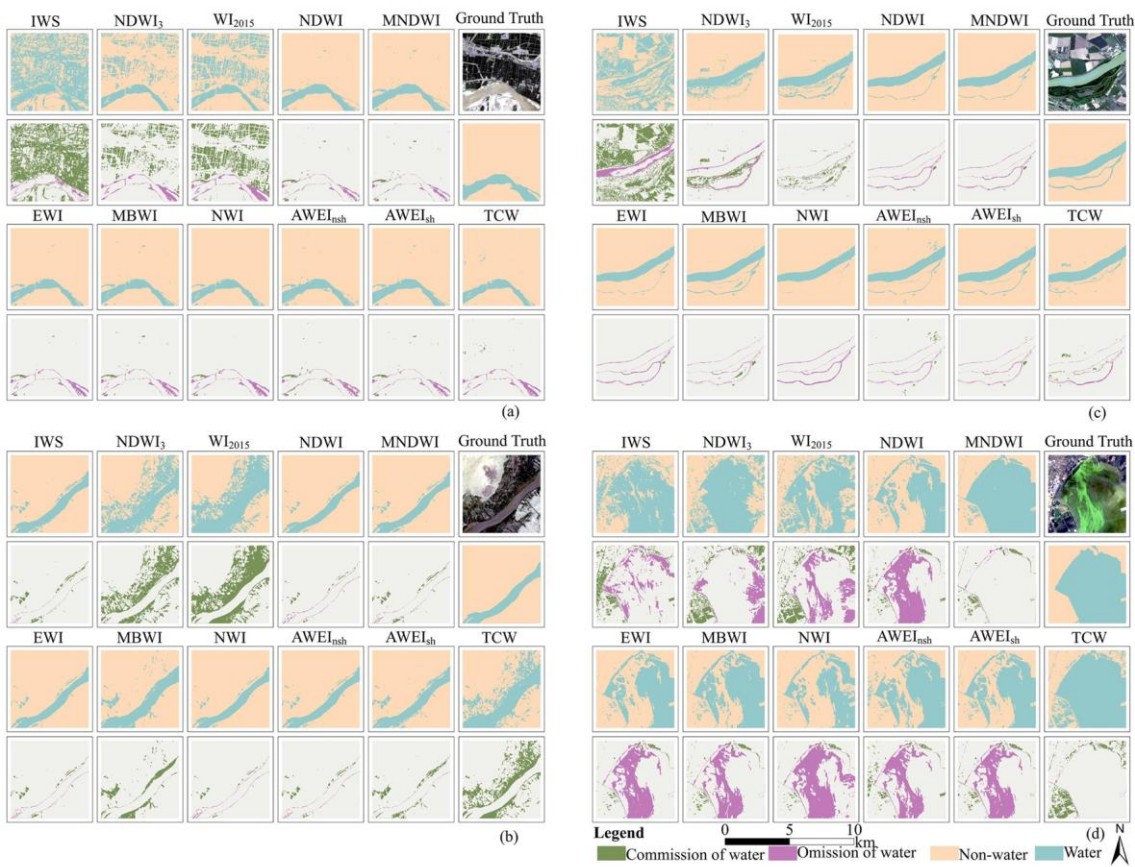

**Figure 7: On the left are the results of turbid water extraction: (a) the Yellow River and (b) the Nile River. On the right are the results of green water extraction: (c) the Danube River and (d) Lake Taihu. All remote sensing images are from Landsat-8 data provided by the GEE platform.**

The presence of salt ions in saltwater induces significant changes in its spectral curve compared with that of freshwater. Within the visible light range, the reflectance gradually diminishes with increasing wavelength, reaching its minimum in the near-infrared band before increasing again in the shortwave infrared band. Notably, the reflectance values for each band are consistently lower than those observed for freshwater: reductions of 17.81% in the blue band, 57.52% in the green band, 91.64% in the red band, 103.75% in the near-infrared band, and 79.46% and 68.33% for both shortwave infrared bands.

This change led to the obvious omission OWIs of the $NDWI_3$, the NDWI, the EWI, and the IWS. The minimum difference between the green band and the near-infrared band underwent a transition from 0.03 to 0.01, thereby leading to the exclusion of NDWI and EWI in saltwater lake extraction (Figure 8.). The ratio between the blue wave band and the first short-wave infrared band is considered a unified entity, and the IWS is transformed into a functional form for analysis. When the ratio between the blue wave band and the first shortwave infrared band falls within the range of 0.27-3.73, IWS also classifies both

waters and other ground objects as positive pixels. However, because, the reflectance of the blue band is three times greater



than that of the first shortwave infrared band in saltwater environments, the IWS is unsuitable for application under such conditions. The reflectance of the first shortwave infrared band of saltwater is 0.004 higher than that of the near-infrared band. This upward trend contradicts that observed in normal water, rendering $NDWI_3$ unsuitable for application in saltwater regions.

As a nonaquatic region with high soil moisture content, the spectral curve of marsh regions is similar to that of water in several aspects. A gradual decrease is observed from the near-infrared to the shortwave infrared bands, with a difference of 0.10. An absorption valley is evident in the shortwave infrared band. Notably, there is a phenomenon where the reflectance values for the blue or green bands surpass those of both the first and second shortwave infrared bands but remain lower than those of the near-infrared band for swamps. The $AWEI_{nsh}$, the $AWEI_{sh}$, the EWI, the IWS, the MBWI, the MNDWI, the TCW, and $WI_{2015}$ all exploit the difference between a blue or green band and a shortwave infrared band. However, in these indexes, the difference between the green band and the near-infrared and shortwave infrared bands is utilized by the EWI and $AWEI_{sh}$. Notably, the weight assigned to the near-infrared band is approximately equal to or slightly less than that assigned to the shortwave infrared band (0.25). This weighting scheme accounts for the enhanced discriminatory effect of these two indexes on swamp water compared with the other ten OWIs (Figure 8.).

In $WI_{2015}$, areas with a soil moisture content greater than 50% during the tectonic process were considered to contain water (Adrian et al., 2016), including some swamps. The NDWI leverages the disparity between green and near-infrared spectral bands to enhance the identification of swamp water. Specifically, the normalized value for the near-infrared and shortwave infrared bands in swamps is 0.34, which contrasts positively with the normalized value of 0.25 observed in clear water. Therefore, in the $NDWI_3$ image, both bright colors cannot effectively distinguish between swamps and water.



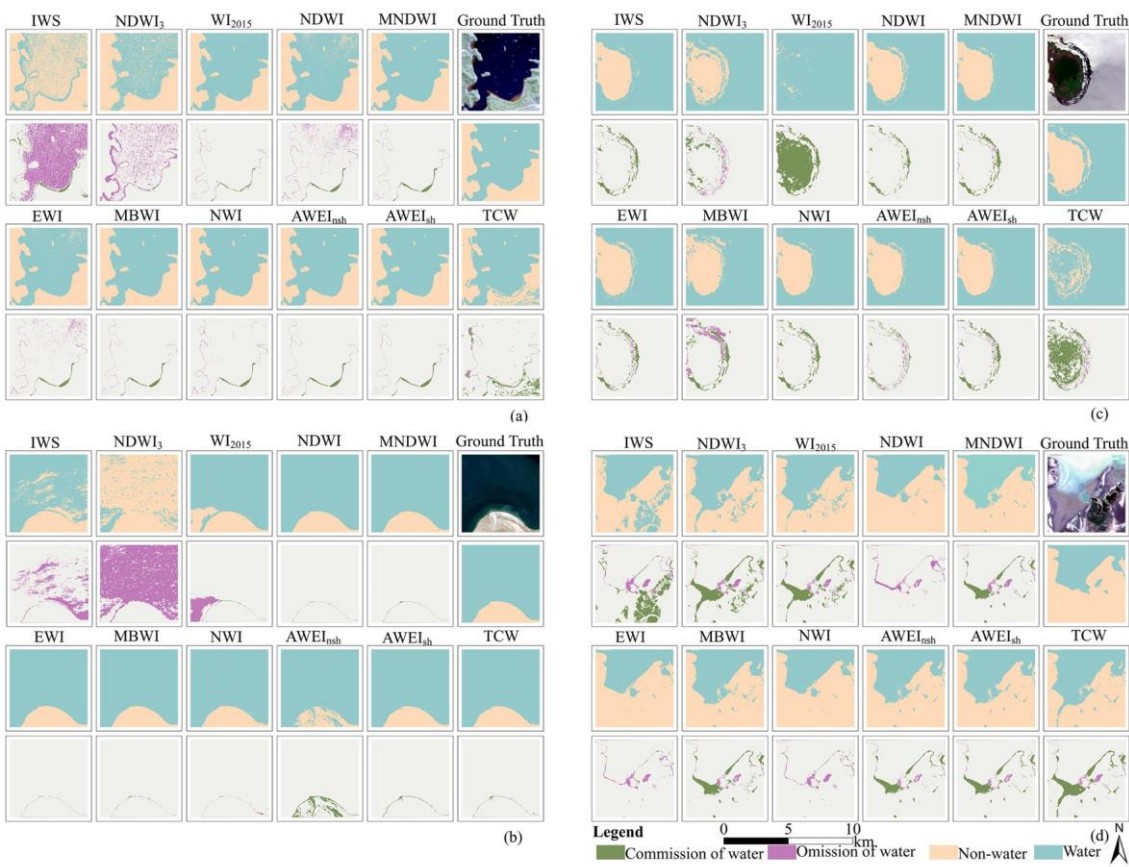

**Figure 8: On the left are the results of saltwater extraction: (a)Lake Eyre and (b)Namtso Lake. On the right are the results of swamp water extraction: (c) Lake Okeechobee and (d) Poyang Lake. All remote sensing images are from Landsat-8 data provided by the GEE platform.**

In areas with water represented by dark pixels, where sunlight is obstructed by surrounding buildings or mountains, the reflection of sunlight on the water surface is diminished, resulting in a darker appearance. The reflectivity of each spectral band in these shaded regions is lower than that of normal waters. Specifically, the blue band experiences a reduction of 46.66%, the green band decreases by 59.04%, and the red band diminishes by 59.48%. Moreover, there is an 11.58% decrease in near-infrared reflectance, a 50.00% decrease in the first short-wave infrared band, and 54.06% decrease in the second shortwave infrared band. Consequently, challenges arise when attempting to accurately extract shaded water due to potential confusion and omission (Figure 9.).

The difference between the mean reflectance of shaded water in the blue band and the near-infrared band is 0.01, and the smallest difference is 0, which leads to the NWI, the $AWEI_{sh}$, and $WI_{2015}$, which use the water index of the difference between the blue band and the near-infrared band, processing the shaded water as dark, as it is confused with background information. The difference between the similar green band and the first shortwave infrared band changes from 0.05 to 0.02 for shaded





water, which results in the omission and over-extraction of shaded waters by the EWI and $AWEI_{nsh}$, which exploit the

difference between the green band and the first shortwave infrared band. When the background object is a building, the ability of the NDWI to distinguish between shaded water and normal water is also compromised because of confusion with the building. This occurs because both the green band and near-infrared band exhibit positive differences in both the building background and water (Mcfeeters, 1996), and a similar phenomenon can be observed in $NDWI_3$.

In shaded water, the ratio between the blue band and the first shortwave infrared band also falls within the range of 0.27-

3.73. Consequently, the IWS categorizes shaded water as non-water features, akin to saltwater.

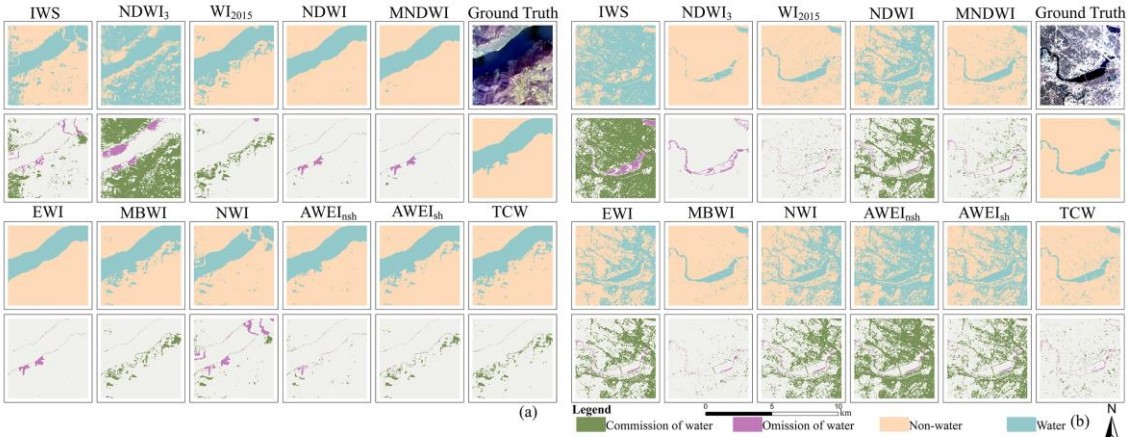

**Figure 9: Extraction results of shaded water: (a) Lake Geneva and (b) the Charles River. All remote sensing images are from Landsat-8 data provided by the GEE platform.**

**4.2 Drawbacks**

In the research process, we attempted to control variables, but the interference of different types of water and different backgrounds was interwoven, including the combination of turbid water and swamp background, the combination of green water and swamp background, and the combination of saltwater and cloud shadows. The excellent performance of the NDWI in identifying swamp water also improves its accuracy in identifying turbid water. However, K-means classification is sensitive

to noise and outliers, and the classification results of complex ground objects are not ideal, which limits the accuracy of classification results to some extent.

**5 Conclusions**

Diversity of water resources across the earth system. The spectral characteristics of different water types vary. The $AWEI_{sh}$, NDWI, and EWI have good separation effects for swamp water. When selecting OWIs for swamp areas, it is advisable to



avoid the shortwave infrared band or reduce its weight. The MBWI demonstrates superior extraction performance for shaded water. The AWEI$_{sh}$ is most suitable for saltwater. When selecting OWIs for saltwater, attention should be given to the increasing trend from the near-infrared band to the short-wavelength infrared band. The MNDWI performs exceptionally well in the extraction of green water. When choosing OWIs for green water, it is essential to consider the increasing reflectance in the near-infrared band. Among the considered indexes, the NDWI emerges as the top performer for extracting turbid water. Both the MNDWI and the MBWI exhibit remarkable stability across diverse water types. The overall accuracy of OWIs constructed through regression and iteration is lower than the average of the 12 OWIs. Sentinel-2 has a higher ability to classify water with higher spatial resolution than Landsat-8. To increase the classification accuracy of OWIs in the future, it is imperative to comprehensively consider the spectral characteristics of different types of water and diverse backgrounds during the construction process. Our work provides theoretical and technical support for a better understanding of the global water system and for rapid and accurate highly automated water mapping.

*Data availability*. The data that support the findings of this study are available from the figshare platform with the doi:10.6084/m9.figshare.25282675.

*Author contributions*. Haifeng Tian designed the study. Yuqing Wang performed the analysis, wrote the manuscript, and made the figures, in direct collaboration with Haifeng Tian and Yaochen Qin. Yijie Ma and Tingsong Gong assisted with data processing. Xueyue Liang, Jie Pei and Li Wang edited and reviewed this article. All authors provided feedback on the manuscript.

*Competing interests*. The authors declare no conflicts of interest.

*Financial support*. This work is supported by the Henan Provincial Science and Technology Research and Development Plan Joint Fund, China (grant no. 222103810029); Open Fund of Key Laboratory of Geospatial Technology for the Middle and Lower Yellow River Regions (Henan University), Ministry of Education (grant no. GTYR202204); the National Natural Science Foundation of China (grant no. 32130066); Xinyang Academy of Ecological Research Open Fund (grant no. 2023XYQN01); Natural Science Foundation of China (grant no. 42301104); Postdoctoral Fellowship Program of CPSF (grant no. GZC20230677); and the major project of the Collaborative Innovation Center on Yellow River Civilization, jointly built by the Henan province and the Ministry of Education (grant no. 2020M19).

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
