# Peer review of "Is It Feasible to Use a Single Remote Sensing Optical Water Index for Rapid Mapping of Water Resources?"

_Natural Hazards and Earth System Sciences, 2024_

## Referee Comment (RC2)

[referee-annotated manuscript omitted]

---

## Author Comment (AC1)

**Response to Referee #1**

Dear reviewer,

Sincerely, thank you for examining the manuscript; these comments have been very helpful in improving this work as well as our subsequent work. I will respond to each of these comments point-by-point in the document below.

Once again, thank you sincerely for your time and dedication to our work.

Best wishes,

Yuqing Wang (on behalf of the author team)

**General comments**

*Question A: lines 53 to 92, where the water mask indexes are explained, should be in the Methods, not in the Introduction.*

**Response A:** Thank you for your comments. We have made this part of the text as an Introduction part, which is to organize and summarize the process of development of existing OWIs. The purpose of this manuscript is to investigate the applicability of existing OWIs to support rapid water resources mapping. Therefore, in the Methods section, we focus on the methods used to conduct the applicability study of OWIs rather than the development process of these OWIs. Therefore, we have included lines 53-92 as part of the Introduction rather than as part of the Methods.

*Question B: There is a section "Method" within the section "Material and Methods". There is no section to provide details on the satellite data that were used, there is only one sentence about it in the "Data" section.*

**Response B:** Thank you for your suggestions, which have been very helpful in enhancing my manuscript. I have modified the Data by adding the corresponding satellite data as shown below.

**Page6, Line138-143:**

Landsat-8 was launched in 2013 and has now been in service for more than a decade, with a revisit period of 16 days, a total of 9 bands, and a spatial resolution of 30m (Barsi et al., 2014).Sentinel-2 was jointly developed by the European Commission and the European Space Agency under the Copernicus program (Berger et al., 2012), and so far has three satellites, Sentinel-2A/B/C. In this study, we use Sentinel-2A, launched in 2015, which has a revisit period of 10 days, 13 bands, and original spatial resolutions of 10m, 20m, and 60m. We then use the OWIs computed by Sentinel-2 to unify the spatial resolution to 10m in GEE.

*Question C: Lines 252-257 should be in the Discussion section.*

**Response C:** Thank you very much for your comments. I have made the appropriate changes in the manuscript. Moved this section to Line 364-368 in the Discussion section.

*Question D: Moreover, English language and grammar require critical revision. There are many grammar errors in the text, including verb conjugations, sentences that are too long or built incorrectly, and misspelled words. In addition, there are portions of the text such as lines 98-101, where one sentence is repeated twice, which indicates that the manuscript was not carefully revised by the authors. The poor language makes it often difficult to understand what the authors mean with some*

*statements.*

**Response D:** I sincerely apologize for my mistake and thank you for reading and checking the manuscript. I read the whole text carefully and checked the English grammar. Verbs, sentence structure, spelling mistakes, etc. were corrected. Again, a sincere apology.

**Specific questions**

*Question 1: The water types are classified as turbid, green, shaded, swamp, and salt waters. However, the criteria for defining these water types are not described.*

**Response 1:** Thank you for your comments. It is difficult to use a quantitative criterion and absolute uniformity for the delineation of different types of water. In this study, we have not given an absolute delineation criterion, and the amount of work required to achieve this uniformity has detracted from the purpose of our work. However, we select different types of water based on their environmental conditions and water quality typicality, and find that different types of water have different spectral characteristics. I add a discussion of this issue to the Discussion section as follows.

**Page18, Line366-368:**

On the other hand, it is difficult to quantitatively standardize the classification of water, such as determining how much sand should be classified as turbid water. However, we observed unique spectral characteristics for different water types in these areas, which are indicative of both water quality and the environment.

*Question 2: There is no definition for the term "surface water", often mentioned in the manuscript.*

**Response 2:** Thank you sincerely for your suggestions. I have revised the manuscript accordingly by adding the definition of surface water.

**Page2, Line38-39:**

Different studies will define surface water slightly differently; in this work, we define surface water as water that is not covered by features other than aquatic plants.

*Question 3: There are no details about the in-loco observations (field surveys) mentioned in line 104. Conversely, according to Fig. 3, model accuracy was tested by comparing outputs of the different models to the output of SWI. If that's the case, what are the field surveys for? Please explain and organize this explanation in the text.*

**Response 3:** Sincere thanks for your questions, they are very helpful in enhancing my manuscript. The field survey was designed to validate the literature representations on the one hand and the manual digitization on the other hand. The original text was inappropriately worded in part. Figure 3 caused some improper understanding that we do not use the SWI identification results as a benchmark against the identification results of other OWIs. I have modified Figure 3 and the text accordingly, as shown below.

**Page5, Line104-105:**

As shown in Figure 1, we selected a total of 10 study areas in this study: the Yellow River, the Nile River, Lake Taihu, the Danube River, Namtso Lake, Lake Eyre, Lake Geneva, the Charles River, Poyang Lake, and Lake Okeechobee.

**Page9, Line185-186:**

The results of the manual digitization were validated against the data from the field survey.

[Figure]

**Figure 1: The overall workflow of this study.**

*Question 4: Equations for 11 water masks, not 12, are shown in Table 1.*

**Response 4:** Thank you for checking. AWEI contains two formulas $AWEI_{sh}$ and $AWEI_{nsh}$, the former is used for shaded areas and the latter is used for non-shaded areas, and they have different uses. Therefore, we study them in their entirety according to the 12 formulas.

*Question 5: Thresholds for water classification, for each index, are not provided. The method used to select this threshold is also not provided.*

**Response 5:** Thank you for your comments. In this study we did not use the thresholding method to partition water from non-water. This is because the determination of the optimal threshold is complex and it is difficult for us to guarantee that the thresholds of all 12 OWIs are optimal. In order to conduct the study more objectively, we chose the K-means classification method in unsupervised classification, which we discuss in the manuscript as follows.

"Classification methods that use OWIs as inputs to classify water and other classes from OWIs can be roughly divided into three categories: supervised classification, unsupervised classification, and threshold methods (Li et al., 2016). Some scholars use the threshold method (Adrian et al., 2016); however, choosing the optimal threshold is very complex and time-consuming (Liu et al., 2023). The Otsu method (1979) is a widely used automatic thresholding method aimed at maximizing interclass variance and minimizing intraclass variance (Du et al., 2016). However, the algorithm does not yield good results for images without bimodal features (Zhou et al., 2015). Supervised classification and its accuracy is depend on the quality of the training samples (Shin et al., 2016). An unsupervised classification algorithm does not rely on training samples and has less subjective interference (Tian et al., 2024). Therefore, we ultimately chose an unsupervised classification method to classify OWI

images to pursue objective evaluation results."

"The K-means classification method is widely employed for unsupervised classification. In the K-means algorithm, cluster analysis is utilized to randomly determine the central locations of clusters and subsequently group the objects that are closest to these centers (Piloyan and Konečný, 2017). Through iterative calculations, the values of each clustering center are updated individually until the optimal clustering outcome is achieved. In this study, the calculation results of twelve OWIs were classified via the K-means method. Specifically, the K-means classification method was employed to partition the images into five categories through ten iterations. Each scene contains approximately five categories: water, vegetation, impervious cover, bare ground, and swamp-and 10 iterations constitute the number of iterations with the highest accuracy that we obtained after repeated experiments. These categories were then aggregated into two overarching groups, 'water' and 'non-water', and the merging process compared true color images, thereby constructing a binary representation."

*Question 6: Paragraph of line 90 should be in the legend of Table 1, since it describes the table content, and the table legend itself is incomplete.*

**Response 6:** Sincerely thank you for your advice, it is very useful in enhancing my manuscript. I have already made the appropriate changes in the manuscript.

**Page3, Line86-91:**

Table 1. Formulas for the optical water index, where $\beta_{blue}$ refers to the reflectance of the blue band (B2 of Landsat-8 and Sentinel-2), $\beta_{green}$ refers to the reflectance of the green band (B3 of Landsat-8 and Sentinel-2), $\beta_{red}$ refers to the reflectance of the red band (B4 of Landsat-8 and Sentinel-2), $\beta_{nir}$ refers to the reflectance of the near-infrared band (B5 of Landsat-8, B8 of Sentinel-2), $\beta_{swir1}$ refers to the reflectance of the first shortwave infrared band (B6 of Landsat-8, B11 of Sentinel-2) and $\beta_{swir2}$ refers to the reflectance of the second shortwave infrared band (B7 of Landsat-8, B12 of Sentinel-2), and $\beta_{vre1}$ refers to the reflectance of the vegetation red edge 1 band (B5 of Sentinel-2).

*Question 7: Line 116-117: in "At present, the water bloom in Taihu Lake is still serious.", please explain what a "serious bloom" is.*

**Response 7:** Sincerely thank you for your suggestion, it makes my manuscript more in-depth. Although the water bloom of Taihu Lake has been improved, there are still very serious water bloom problems, the biodiversity of Taihu Lake is reduced, the drinking water safety of the surrounding residents is threatened, and hundreds of thousands of people are involved in the salvage of water bloom every year. I have made corresponding changes in the manuscript.

**Page5, Line113-116:**

At present, the water bloom in Taihu Lake remains severe (Wang et al., 2020). This has led to a reduction in the biodiversity of Taihu Lake, threatened the drinking water safety of surrounding residents, and resulted in hundreds of thousands of people being involved in water bloom salvage efforts every year.

*Question 8: Legends of all figures and tables are too short and do not explain what the content is.*

**Response 8:** Thank you for your comments. I have made the appropriate changes in the manuscript.

*Question 9: Section 4.1: What phenomenon?*

**Response 9:** Sincere thanks for your question, which improve the quality of my manuscript and guide

subsequent writing. I have revised the manuscript accordingly, as follows.

**Page14, Line267:**

**4.1 Reasons for omissions and confusion**

**Reference**

Adrian, Fisher, Neil, Flood, Tim, and Danaher: Comparing Landsat water index methods for automated water classification in eastern Australia, Remote Sens. Environ., 175, 167-182, https://doi.org/10.1016/j.rse.2015.12.055, 2016.

Barsi, J., Lee, K., Kvaran, G., Markham, B., and Pedelty, J.: The Spectral Response of the Landsat-8 Operational Land Imager, Remote Sens., 6, 10232-10251, https://doi.org/10.3390/rs61010232, 2014.

Berger, M., Moreno, J., Johannessen, J. A., Levelt, P. F., and Hanssen, R. F.: ESA's sentinel missions in support of Earth system science, Remote Sens. Environ., 120, 84-90, https://doi.org/10.1016/j.rse.2011.07.023, 2012.

Du, Y., Zhang, Y., Ling, F., Wang, Q., Li, W., and Li, X.: Water Bodies' Mapping from Sentinel-2 Imagery with Modified Normalized Difference Water Index at 10-m Spatial Resolution Produced by Sharpening the SWIR Band, Remote Sens., 8, https://doi.org/10.3390/rs8040354, 2016.

Li, W., Qin, Y., Sun, Y., Huang, H., Ling, F., Tian, L., and Ding, Y.: Estimating the relationship between dam water level and surface water area for the Danjiangkou Reservoir using Landsat remote sensing images, Remote Sens. Lett., 7, 121-130, https://doi.org/10.1080/2150704X.2015.1117151 2016.

Liu, S., Wu, Y., Zhang, G., Lin, N., and Liu, Z.: Comparing Water Indices for Landsat Data for Automated Surface Water Body Extraction under Complex Ground Background: A Case Study in Jilin Province, Remote Sens., 15, 1678, https://doi.org/10.3390/rs15061678, 2023.

Shin, J. I., Kim, I. J., and Kim, D. W.: Accuracy Assessment of Supervised Classification using Training Samples Acquired by a Field Spectroradiometer: A Case Study for Kumnam-myun, Sejong City, Journal of the Korean Society for Geo-spatial Information Science, 24, 121-128, https://doi.org/10.7319/kogsis.2016.24.1.121, 2016.

Tian, H., Wang, S., Wu, F., Qin, Y., Zhang, X., Wang, L., Pei, J., Liu, J., and Yang, M.: Comparing the potentials of the different canola flower indices for canola mapping based on Landsat 9 images, All Earth, 36, 1-13, https://doi.org/10.1080/27669645.2023.2291216, 2024.

Zhou, C., Tian, L., Zhao, H., and Zhao, K.: A method of Two-Dimensional Otsu image threshold segmentation based on improved Firefly Algorithm, 1420-1424 pp., https://doi.org/10.1109/CYBER.2015.7288151, 2015.

---

## Author Comment (AC2)

**Response to Community Comments**

Dear Chenxi Shi,

Sincerely thank you for your comments on the manuscript, they are very helpful in this endeavor. I will respond to each of these comments point-by-point in the document below.

Thanks again for your support of this process.

Best wishes,

Yuqing Wang (on behalf of the author team)

*Question 1: P2, L32: The combination of OWIs and SAR for flood mapping is indeed common, but what is the difference between the two and what are the advantages of OWIs?*

**Response 1:** Sincerely thank you for your comments, it has been very helpful in improving the quality of my manuscript. The difference between the two is that SAR's recognition of water is not as ideally accurate as optical images, which is an advantage of OWIs. I have modified the manuscript accordingly as follows.

**Page32, Line34:**

However, SAR struggles to identify water with the same accuracy as optical images (Cho and Qi, 2022).

*Question 2: P3, L83, 84: This paragraph could be merged with the previous one.*

**Response 2:** Thanks to your suggestion, I have revised the manuscript accordingly by merging these two paragraphs.

*Question 3: P8, Fig. 2(b) Swamp and Vegetation colors are not easily distinguishable and need to be modified.*

**Response 3:** Sincerely thank you for your suggestions. I have made changes to the images in the manuscript as follows.

**Page8, Line171-174:**

[Figure]

**Figure 1: Spectral figures of reflectance of various features. (a) Spectral curves for 5 types of water. (b) Spectral curves from swamps, buildings, vegetation, and soil. The center line is the**

**mean value of the reflectance and the surrounding bands are the standard deviation of the reflectance.**

*Question 4: P9, once again, Fig3 has some colors very close together, make changes.*

**Response 4:** Thank you sincerely for reading the manuscript. I made changes to the Figure3 accordingly.

**Page9, Line183:**

[Figure]

**Figure 2: The overall workflow of this study.**

*Question 5: The Discussion includes "Explanation of the phenomenon" and "Drawbacks"; the author should add a related parties with future research.*

**Response 5:** Sincere thanks for your suggestions, which have enriched the content of my manuscript. Regarding future research, I think the nodes of change in the applicability of OWIs can be explored in conjunction with water quality studies. The results of this paper can also be validated in the context of floods and droughts. I have modified the manuscript accordingly as below.

**Page19-20, Line380-384:**

**4.3 Prospect**

Future research could focus on studying water quality changes and identifying specific water quality parameters where the applicability of OWIs varies across different contexts. The goal is to further enhance the automation of water mapping in extreme situations. Additionally, exploring OWI applicability in flood and drought disasters would validate the findings of this study and contribute to the advancement of OWI development.

*Question 6: The Conclusions are short and could usefully be augmented to emphasize the significance of the work.*

**Response 6:** Thank you for your suggestions, I recognize your point of view. I have made further additions to the manuscript.

**Page20, Line398-399:**

At the same time, this work can provide experience in timely and accurate monitoring of water resources in the event of disasters, saving time in the selection of methods.

---

## Author Comment (AC3)

**Response to Referee #2**

Dear reviewer,

I am very happy to have received your comments. I sincerely appreciate the time you spent reading the manuscript and the comments were very helpful. I will respond to each of these comments point-by-point in the document below.

Best wishes,

Yuqing Wang (on behalf of the author team)

***Question 1:*** *Surface waters? Rivers...lakes...not underwater water resource.*

**Response 1:** Thank you for your suggestion, I have made a change to the title and the whole text accordingly.

**Page1, Line1-2:**

Is It Feasible to Use a Single Remote Sensing Optical Water Index for Rapid Mapping of Surface Water?

**Page1, Line15-17:**

Surface water is an important component of the earth's system, and the frequent occurrence of floods and droughts in the context of current climate change makes rapid and accurate monitoring of these resources even more essential.

**Page1, Line24-25:**

Our work provides prior experience for fast and accurate surface water mapping in case of floods or droughts.

**Page5, Line113-115:**

In this work, we aim to quantify the strengths and weaknesses of the twelve OWIs, which can provide technical and theoretical support for better surface water monitoring and rapid mapping in case of related disasters.

***Question 2: Please provide some of the metrics to show the quality of each index applied to each water body.***

***Also, each of the water groups (turbid...green...etc) are not a standard nomenclature...which can cause some misinterpretation...***

**Response 2:** Sincerely thank you for your comment, this issue is indeed our shortcoming, as mentioned by the first reviewer. In this study, we have not given an absolute delineation criterion, and the amount of work required to achieve this uniformity has detracted from the purpose of our work. However, we select different types of water based on their environmental conditions and water quality typicality, and find that different types of water have different spectral characteristics. I add a discussion of this issue to the Discussion section as follows.

**Page18, Line366-368:**

On the other hand, it is difficult to quantitatively standardize the classification of water, such as determining how much sand should be classified as turbid water. However, we observed unique spectral characteristics for different water types in these areas, which are indicative of both water quality and the environment.

***Question 3:*** *I suggest rewrinting this sentence expanding to some of the differences between optical and radar remote sensing...to contextualize the use of indexes...*

**Response 3:** Sincerely thank you for your advice and I very much recognize your point of view. The difference between microwave and optical hydrography is that microwave hydrography is not as accurate as optical hydrography. Changes have been made in the manuscript accordingly.

**Page2, Line33-35:**

It is also widely used in hazard mapping in conjunction with Synthetic Aperture Radar (SAR) because the most serious problem with optical imaging is the influence of clouds (Psomiadis et al., 2020). However, SAR struggles to identify water with the same accuracy as optical images (Cho and Qi, 2022).

***Question 4:*** *this sentence looks confusing to me. Is this a "methodology"?*

**Response 4:** Sincerely thank you for your question. The sentence did seem a bit confusing and I have removed it from the manuscript.

***Question 5:*** *Please, provide a reference for that.*

**Response 5:** Sincerely thank you for your comments, references have been added.

**Page2, Line52-53:**

Most OWI methods are constructed on the basis of Landsat series images(Adrian et al., 2016; Liu et al., 2016).

***Question 6:*** *Please provide more references for that.*

**Response 6:** Sincerely thank you for your comments, references have been added.

**Page2, Line55-56:**

The emergence of Sentinel-2 has significantly enhanced the monitoring of water areas via remote sensing owing to its higher spatial resolution and increased spectral bands (Zeng et al., 2022; Jiang et al., 2021).

***Question 7:*** *This sentence looks confusing. something wrong here...It looks something is wrong here...*

**Response 7:** Sincerely thank you for checking and I apologize for my mistakes. I made the correct changes in the manuscript and again sincerely apologize.

**Page3, Line72-75:**

They concluded that a single-band difference would not be sufficient to eliminate all sources of interference. To address this issue, they introduced the Enhanced Water Index (EWI), which combines the NDWI and MNDWI. This is the first study in which the effects of a dry river on water extraction have been reported.

***Question 8:*** *You should cite the table before showing it in the text...*

**Response 8:** Sincerely thank you for your comment, changes have been made accordingly. Use the cross-reference function after adding a caption to a table.

**Page5, Line107:**

Although many OWIs are available (Table 1), they can be deceptive.

*Question 9: It looks similar...*

**Response 9:** Sincerely apologize for my mistake. I have scrutinized the manuscript and made corrections for this and similar errors. I promise that similar problems will never occur in a further revised manuscript. Sincerely thank you for your comments.

*Question 10: I believe there is a need for a table displaying the main characteristics of each lake, as well as some of the constituents or limnological features, to enhance the applicability range of each index tested.*

**Response 10:** Sincerely thank you for your comments, which have been very helpful in improving the quality of my manuscript. I agree with you that the addition of a table does make the presentation clearer. However, rather than focusing on the full range of hydrologic characteristics of these waters, I believe the content of the table should emphasize their typicality as study areas. This would make the center of the manuscript more prominent.

**Page6, Line153-154:**

**Table 1: Correspondence table between the study area and water types.**

| Water type | Turbid water | Green water | Salt water | Shaded water | Swamp water |
|---|---|---|---|---|---|
| Name of water area | Yellow River Blue Nile | Danube River Taihu Lake | Namtso Lake Lake Eyre | Lake Geneva Charles River | Poyang Lake Lake Okeechobee |

*Question 11: Please, explain what type of grond-truth observation is this...and also include it in a table ...*

**Response 11:** Thank you sincerely for your comment. The presentation here is indeed not very clear. Our field data are meant to be validated against the manually digitized results from the accuracy validation. They are no longer presented in tables because they are very voluminous. I have made the corresponding changes in the manuscript as follows.

**Page7, Line164:**

We also combined ground-truth observation data as validation data for accuracy validation.

**Page11, Line208-209:**

The results of the manual digitization were validated against the data from the field survey.

*Question 12: The previous table will help even more to understand those characteristics...*

**Response 12:** Sincere thanks for your comment. I believe this is a combined comment with comment #10 and have made the appropriate changes in the manuscript.

*Question 13: The K-Means is applied before the indexes? this should be included in the flowchart...*

**Response 13:** Sincerely thank you for your question, it has been very helpful in enhancing my manuscript as well as subsequent writing. K-means comes after exponential computation. Our description in the manuscript is shown below. "In this study, the results of calculating twelve OWIs were classified using the K-means method." The original flowchart in the manuscript was really not clear in its presentation, and I have modified it accordingly, as shown below.

**Page10, Line203:**

[Figure]

**Figure 1: The overall workflow of this study.**

*Question 14: Please, define what is an "unclear water"...*

**Response 14:** Thank you sincerely for your comment. The expression "unclear water" is inappropriate, I wanted to express that there are many different types of water in nature and not all of them are ideally clear (the spectral curve decreases in reflectance as the wavelength increases). In addition to these changes, of course, we still need to define "unclear water". We define unclear water as one that has a significant change in its spectral profile from that of primary source water. I have modified the manuscript accordingly.

**Page8, Line177-178:**

These OWIs behave differently in recognizing water in nature, which are diverse and not always ideally clear (the spectral curve decreases in reflectance as the wavelength increases).

**Page8, Line186-187:**

We define unclear water as one that has a significant change in its spectral profile from that of primary source water.

*Question 15: Please, take a look in the meaning of "signifies" in this sentence...*

**Response 15:** Sincerely thank you for reading, your suggestions have been very helpful in improving the quality of my manuscript. The "signifies" here is indeed an error that needs to be corrected, and I have made the appropriate changes to the manuscript.

**Page8, Line184-186:**

*where $\delta$ represents the percentage increase or decrease in each band for different types of water, $\beta_{unclear}$ denotes the reflectance of each band of the unclear water, and $\beta_{clear}$ denotes the reflectance of each spectral band of water in the first-class water source protection area.*

*Question 16: In situ?*

**Response 16:** Thank you sincerely for your question. The sampling point data here are derived from remote sensing imagery, and I have made changes in the manuscript accordingly.

**Page8, Line187-189:**

Spectral samples of clear water were obtained from satellite imagery of the Danjiangkou Reservoir, the largest artificial freshwater lake in Asia and a national first-class water source protection area (Pan et al., 2021).

*Question 17: Where are those samples come from? I mean, the spectral signature...it looks very high spectral resolution.*

**Response 17:** Sincere thanks for your comments, these samples are from Landsat images from our ten study areas. Our text in the manuscript reads as follows. "The spectral data of the remaining five water types and various background features were acquired from 10 study areas via the Landsat-8 dataset (Figure 2)." I'm guessing your problem is that green water has a very high reflectance in the near infrared band. This is because the sampling point of Lake Taihu is included here, and the problem of water bloom in Lake Taihu is very serious, and we have repeatedly verified that the spectral curve of this region is indeed so. We have provided additional information in the manuscript, taking into account the comments of the first reviewer.

**Page6, Line132-134:**

At present, the water bloom in Taihu Lake is remains severe (Wang et al., 2020). This has led to a reduction in the biodiversity of Taihu Lake, threatened the drinking water safety of surrounding residents, and resulted in hundreds of thousands of people being involved in water bloom salvage efforts every year.

*Question 18: ROIs?*

**Response 18:** Thank you for your suggestion. It is indeed ROIs and has been changed accordingly in the manuscript.

**Page11, Line209-210:**

As accuracy verification results can be significantly influenced by the selection of validation samples (Adrian et al., 2016), we adopted a systematic approach by selecting 100 ROIs, each measuring 6 km×6 km.

*Question 19: The colors on Figure 4 should guide the reader for the results...*

**Response 19:** Sincerely thank you for reading. Warmer colors in Figure 4 represent higher recognition accuracy, and cooler colors represent worse recognition accuracy, and I have made changes in the manuscript accordingly.

**Page11, Line230-231:**

Warmer colors in Figure 4 represent higher recognition accuracy, and cooler colors represent worse recognition accuracy.

*Question 20: I would separate the discussion aiming on each of the "classes" previously stated: Turbid, Saltwater, Shadead Water, Green Water and Swamp.*

**Response 20:** Sincerely thank you for your suggestion, which made my discussion section clearer. I have made the appropriate changes in the manuscript.

*Question 21: How did you calculated that?*

**Response 21:** Your comments are sincerely appreciated. The variations in the individual bands are relative to the data sampled from the remote sensing images at Danjiangkou Reservoir. I have

discussed this in detail in the Methods section of the manuscript. "These OWIs behave differently in recognizing water in nature, which are diverse and not always ideally clear (the spectral curve decreases in reflectance as the wavelength increases). We quantified spectral variations in different water types by calculating the percentage change relative to clear reference water across various bands in five categories: turbid, green, shaded, swamp, and saltwater. The calculation formula is as follows:

$$\delta = \frac{\beta_{unclear} - \beta_{clear}}{\beta_{clear}} \times 100\%$$

where $\delta$ represents the percentage increase or decrease in each band for different types of water, $\beta_{unclear}$ denotes the reflectance of each band of the unclear water, and $\beta_{clear}$ denotes the reflectance of each spectral band of water in the first-class water source protection area. We define unclear water as one that has a significant change in its spectral profile from that of primary source water. Spectral samples of clear water were obtained from satellite imagery of the Danjiangkou Reservoir, the largest artificial freshwater lake in Asia and a national first-class water source protection area (Pan et al., 2021)."

*Question 22: Figures are too small and hard to understand...*
**Response 22:** I very much recognize your point of view and have modified all such images in the discussion section. I have copied one of them as follows.
**Page17, Line330-333:**

[Figure]

**Figure 2: Recognition of turbid water by OWIs. (a) the Yellow River and (b) the Nile River. All remote sensing images are from Landsat-8 data provided by the GEE platform.**

*Question 23: ???? Not clear for me...*

**Response 23:** Sincerely, we thank you for reading it, and we have revised the manuscript accordingly, as follows.

**Page30, Line483-484:**

The diversity of surface water across the earth system results in varying spectral characteristics for

different water types.

**Reference**

Adrian, Fisher, Neil, Flood, Tim, and Danaher: Comparing Landsat water index methods for automated water classification in eastern Australia, Remote Sens. Environ., 175, 167-182, https://doi.org/10.1016/j.rse.2015.12.055, 2016.

Cho, M. S. and Qi, J.: Characterization of the impacts of hydro-dams on wetland inundations in Southeast Asia, Sci. Total Environ., 864, 160941, https://doi.org/10.1016/j.scitotenv.2022.160941, 2022.

Jiang, W., Ni, Y., Pang, Z., Li, X., Ju, H., He, G., Lv, J., Yang, K., Fu, J., and Qin, X.: An Effective Water Body Extraction Method with New Water Index for Sentinel-2 Imagery, Water, 13, 1647, https://doi.org/10.3390/w13121647, 2021.

Liu, Z., Yao, Z., and Wang, R.: Assessing methods of identifying open water bodies using Landsat 8 OLI imagery, Environmental Earth Sciences, 75, 10.1007/s12665-016-5686-2, 2016.

Pan, X., Lin, L., Zhang, S., Zhai, W., Tao, J., and Li, D.: Composition and Distribution Characteristics of Microplastics in Danjiangkou Reservoir and Its Tributaries, Environmental Science, 42, 1372-1379, https://doi.org/10.13227/j.hjkx.202006123, 2021.

Psomiadis, E., Diakakis, M., and Soulis, K. X.: Combining SAR and Optical Earth Observation with Hydraulic Simulation for Flood Mapping and Impact Assessment, https://doi.org/10.3390/rs12233980, 2020.

Wang, S., Li, J., Zhang, B., Lee, Z., Spyrakos, E., Feng, L., Liu, C., Zhao, H., Wu, Y., Zhu, L., Jia, L., Wan, W., Zhang, F., Shen, Q., Tyler, A. N., and Zhang, X.: Changes of water clarity in large lakes and reservoirs across China observed from long-term MODIS, Remote Sens. Environ., 247, 111949, https://doi.org/10.1016/j.rse.2020.111949, 2020.

Zeng, Y., Hao, D., Huete, A., Dechant, B., Berry, J., Chen, J., Joanna, J., Frankenberg, C., Bond-Lamberty, B., Ryu, Y., Xiao, J., Asrar, G., and Chen, M.: Optical vegetation indices for monitoring terrestrial ecosystems globally, Nat. Rev. Earth Environ., 3, https://doi.org/10.1038/s43017-022-00298-5, 2022.